# Effect of Two Nutritional Strategies to Balance Energy and Protein Supply in Fattening Heifers on Performance, Ruminal Metabolism, and Carcass Characteristics

**DOI:** 10.3390/ani10050852

**Published:** 2020-05-14

**Authors:** Rodrigo A. Arias, Gonzalo Guajardo, Stefan Kunick, Christian Alvarado-Gilis, Juan Pablo Keim

**Affiliations:** 1Instituto de Producción Animal, Universidad Austral de Chile, Valdivia-Chile, Valdivia 5090000, Chile; calvarado@uach.cl (C.A.-G.); juan.keim@uach.cl (J.P.K.); 2Centro de Investigación de Suelos Volcánicos, Universidad Austral de Chile, Valdivia 5090000, Chile; 3Escuela de Graduados, Facultad de Ciencias Agrarias y Alimentarias, Universidad Austral de Chile, Valdivia 5090000, Chile; gonzalo.guajardo.c@gmail.com (G.G.); stefan93_@live.cl (S.K.)

**Keywords:** nitrogen efficiency, synchrony index, metabolizable protein system

## Abstract

**Simple Summary:**

Beef production has been under strong scrutiny during the last half-century. First, because of its supposed negative impact on human health, and more recently, due to the negative impact on the environment, mainly from nitrogen and greenhouse gases. We conducted an experiment to assess the effects of a diet formulated based on the metabolizable protein system and synchronicity between energy and protein, on the nitrogen losses to the environment, performance, and carcass characteristics of the fattening heifers. Our results show that a diet combining a high synchrony index with the metabolizable protein system increases the nitrogen use efficiency without negatively affecting animal performance or carcass characteristics compared to heifers fed a diet without a balanced protein but with a high synchrony index.

**Abstract:**

Latin America is an important contributor to the worldwide beef business, but in general, there are limited studies considering strategies to reduce nitrogen contamination in their production systems. The study’s goal was to assess the effect of two nutritional strategies to balance energy and protein supply in fattening heifers on performance, ruminal metabolism, and carcass characteristics. A total of 24 crossbred heifers (initial body weight ’BW ’of 372 ± 36 kg) were used to create two blocks (based on live weight) of two pens each, that were equipped with individual feeders. Within each block, half of the animals were assigned to a diet based on tabular Crude Protein (CP) requirements denominated Crude Protein Diet ‘CPD’ but without a ruminal degradable protein balance. The other half received a diet denominated Metabolizable Protein Diet ‘MPD’, formulated with the metabolizable protein system, balanced for the ruminal degradable protein. Both diets had the same ingredients and as well as similar synchrony indexes (0.80 and 0.83, respectively). For nitrogen concentration in feces and urine as well as microbial crude protein synthesis, a total of 12 heifers (three per pen) were randomly selected to collect samples. The dataset was analyzed as a randomized complete block design with a 5% significance. No diet × time interaction was observed for Average Daily Gain ’ADG’ (*p =* 0.89), but there was an effect of the time on ADG (*p ≤* 0.001). No differences were observed neither for final weight, dry matter intake ’DMI’, and feed conversion rate (*p >* 0.05). Heifers fed with CPD showed greater cold carcass weight (*p =* 0.041), but without differences in ribeye area, backfat thickness, pH, dressing %, and marbling (*p >* 0.05). Differences between diets were observed for the in vitro parameters as well as for the Total Volatile Fatty Acids ’VFA’ and NH_3_ (*p <* 0.05). Total N concentrations (urine + feces) of heifers fed with MDP was lower than in those fed with the CPD (*p <* 0.01), but no differences were observed in microbial protein, purine derivatives, and creatinine (*p >* 0.05). We conclude that the combination of synchrony and the metabolizable protein system achieve greater efficiency in the use of nitrogen, without negatively affecting animals’ performance or the quality of the carcass.

## 1. Introduction

In many countries, traditional beef production systems have a low level of intensification. Though, the implementation of new technologies in herd management, such as the type and form of feeding, would have a positive response in productive efficiency, which directly impacts the final utility of the producer [1]. Nevertheless, due to the growing world beef demand in conjunction with more environmental restrictions, it is expected that the increase in production is mainly due to more intensification [2]. This new scenario should aim not only to achieve more beef and of better quality but also improve animal welfare and, at the same time, cause minimum environmental impact. However, there is a risk that greater intensification is a source of environmental pollution since it has been associated with highly intensive animal production systems [3]. This is mainly due to nitrogen (N) and phosphorus (P), which pollute groundwater, surface water, and air. The same authors detail that the main reasons for contamination are excretions (urine and feces) that are discharged into surface waters, or by volatilization of N in the form of NH_3_. It also has been suggested that new feeding strategies should be adopted to improve the use of N and reduce its losses. In this sense, Klemesrud et al. [4] state that satisfying the animal’s amino acid requirements, without falling into excess protein in the rations, it allows to reduce N excretions. Thus, adequate management of nutrition should not only allow digesting and utilizing the fiber from forages and promote microbial protein synthesis but also reduce N losses at the ruminal level, which may be excreted to the environment. Farmers and nutritionists should try to maximize microbial protein production in the rumen because it has been established that microbial protein can provide from 50 to 100% of the metabolizable protein (MP) required by the animal, which depends on the amount of ruminal degradable protein and fermentable carbohydrates in the diet [5].

Owen and Sapienza [6] pointed out that to achieve maximum efficiency and production of microbial protein, factors that affect its synthesis should be considered, such as the availability of energy, ammonium or other N sources, minerals, vitamins, and factors related to animal growth. The synchrony between energy and degradable protein in the rumen is a very important factor in the optimization of microbial protein synthesis [7]. Therefore, not only the contribution or availability of energy and nitrogen sources at the ruminal level should be necessary [8,9], but also synchrony [10]. Other authors consider that the amount or availability of energy in the rumen, in a certain nitrogen range, is usually more important than synchrony as such for the synthesis of microbial protein [9,11]. According to Sinclair [12], ruminal synchrony would improve the rumen fermentation, digestibility of nutrients, and the microorganism’s population [13], as well as bacterial protein synthesis, nitrogen retention, the animal’s productive response, and consequently, the N use efficiency (NUE) [14].

Ruminal synchrony could be affected by different factors such as diet, which is the main determinant of the quantity and quality of nutrients supplied to microorganisms [10]. Mention has also been made to the chemical composition of the feedstuffs [15], the ingredients used in the ration [16], grains processing [17], and the degradation characteristics of organic matter and protein in food [13,18]. Other factors related to the animal, such as health, nutritional requirements, and the interaction with the type of management and the environment, also alter the use of nutrients, causing ruminal asynchrony [10]. As mentioned, the synchrony between energy and protein available in the rumen theoretically should allow more efficient use of nutrients, improving the production of microbial products to increase the supply of nutrients, and thus improve the animal’s performance. However, after years of studies and analysis of this concept, it has not been fully demonstrated in practice. Likewise, asynchronous nutrient diets have generated results as good or higher than synchronous diets in growing beef and dairy cattle in confinement [10]. Duarte et al. [19] evaluated the effects of different levels of energy and non-degradable protein in the rumen on consumption, growth, carcass characteristics, and meat quality in fattening heifers. The authors concluded that there was a higher average daily gain (ADG) with high levels of rumen undegradable protein (RUP). However, levels of RUP had no effect on carcass characteristics and composition, but there was an effect on the energy level with a greater area of the eye of the back (longissimus muscle).

At present, the MP system widely used by beef nutritionists, predicts protein requirements accurately, being effective in providing protein levels even at or near the predicted requirements, ensuring good animal performance [20]. Erickson et al. [21] evaluated the effect of the use of diets balanced with the conventional method (based on CP) vs. others based on the MP system in fattening of steers and calves under highly intensive systems in the USA. They determined that there were no differences in the animal’s productive responses. However, the excretion of N was reduced with experimental diets. Nowadays, there is limited information about the implementation of the MP system in the South American beef production systems. Most of these production systems are much less intensive than North American feedlots, such as those studied by Erickson et al. [21]. In this context, we hypothesize that a diet formulated with the MP system will decrease the environmental impact by reducing nitrogen losses to the environment, without negatively affecting in vitro rumen fermentation, the productive performance and the carcass characteristics of the fattening heifers.

## 2. Materials and Methods

All the procedures, including animal care and handling procedures, followed national legislation (Law No. 20,380 on Protection of Animals; Decree No. 29 about regulation on the protection of animals during their industrial production, their commercialization and in other areas to hold animals), whose application is supervised by the National Service of Agriculture and Livestock (SAG), the competent authority in this matter.

### 2.1. Animals and Facilities

The study was conducted at the Austral Agricultural Research Station (EEAA) of the Universidad Austral de Chile, located 8 km North of Valdivia, Los Ríos region (39° 46′28″ S 73° 14′11″W), from September 6th, 2016 until December 6th, 2016. A total of 24 crossbred heifers (Hereford × Angus) with an average initial BW of 372 ± 36 kg were used (68% of mature weight). All heifers were dewormed with 1% ivermectin for the control of gastrointestinal parasites. Upon arrival, animals were grouped based on their initial live weight conforming two blocks (light: 345 ± 23.32 kg and heavy: 399 ± 25.10 kg) and were located in four pens that were randomly assigned to each block. Each block consisted of two pens with a space allowance of 26.25 m^2^ per head. In addition, each pen was equipped with six individual semi-automatic feeders (American Calan Inc., Northwood, NH, USA), which opened through an electronic key that hangs on the neck of each animal. This allowed the control of individual food intake, considering each animal as an experimental and observational unit. Feeders were under a shed to protect feeds from rain, with a cement surface (platform) of 3.5 m wide. Cattle had access to water through two shared drinking fountains of 600 L, each with a float system to ensure their filling, each located between two adjacent pens.

Heifers were weighed approximately every 20 d with an electronic scale (Iconix FX31, Iconix NZ Ltd., Palmerston North, New Zealand). The weighing was performed during the morning before the feed was supplied to avoid variations due to gastrointestinal content. Average daily weight gain (ADG) was estimated at intervals corresponding to weighing dates. Additionally, feed conversion of the whole trial was determined by dividing the total amount of food consumed by the total kg of body live weight (LW) gained. Finally, the LW of heifers was used to calculate the synthesis of microbial protein.

There was a training period of approximately 20 d that was carried out so that the animals got used to the facilities and the feeders, as was previously described by Arias et al. [22]. During this time, heifers were fed ad libitum with a 100% pasture haylage and water. Animals were fed once a day (09:00 to 11:00 am). The ingredients were weighed and mixed daily before supplying the diet in each feeder, and orts were collected and weighed individually (per feeder) once a week. Simultaneously, the preference of each animal for the feeders was observed in order to assign the respective key. Finally, once assigned and the keys placed on the neck of the heifers, the electrical system was connected to block them and, the experimental phase was started, which took place over a period of 64 d. Animals were fed once a day (09:00 to 11:00 am).

### 2.2. Dietary Treatments and Nutrient Concentration Analyses

The chemical composition of feed ingredients and their inclusion in the dietary treatments are reported in Table 1 and Table 2. The animals within each block were randomly assigned to one of two dietary nutritional strategies. Diets were formulated to be isoenergetic. Each diet was designed according to the ruminal degradation parameters of their respective ingredients in order to obtain similar synchrony between energy and protein in the rumen based on the synchrony index (SI) proposed by Sinclair et al. [12]. The SI of both diets was calculated based on the ruminal degradation parameters of OM and CP of each ingredient of the diets adopting the methodology of Verbič et al. [23]. Both diets were calculated to obtain an ADG of 1.0 kg. The first treatment was formulated based on heifers CP requirements and was defined as the ‘Crude Protein diet’ (CPD). It was designed to provide the amount of daily metabolizable energy and crude protein needed for heifers with an SI = 0.80. This diet was analyzed using the Beef Cattle Nutrient Requirement Model (BCNRM) software [24], contrasting the values obtained in the balance for predicted ruminal degradable protein (RDP). According to the National Academies of Sciences, Engineering, and Medicine (NASEM) [24], there was an imbalance of the RDP supply (+240 g d^−1^) since it was calculated based on the CP system. The second dietary treatment ‘Metabolizable Protein diet’ (MPD) was formulated by using the MP system, providing an adequate supply of ruminal undegradable protein (RUP; −10 g d^−1^) and with a similar SI than the CPD (0.83).

Dry matter content was measured by weighing the samples before and after drying with a forced-air oven, initially at 60 °C for 48 h, and then at 105 °C for 12 h. The CP concentration was determined by combustion (Leco Model FP-428, Leco Corporation, St Joseph, MI, USA) based on the DUMAS method (N × 6.25), digestible organic matter on a dry matter basis (DOMD) was measured according to Tilley and Terry [25], neutral detergent fiber (aNDF) was measured by using a heat-stable amylase [26], and ash and ether extract (EE) were analyzed according to AOAC [27] (Methods ID 942.05 and ID 920.39 for ash and EE, respectively).

### 2.3. Organic Matter and Crude Protein in Situ Degradation Parameters of Feed Ingredients

Food samples were lyophilized and grounded through a 5 mm sieve. Polyester bags of 10 × 20 cm (40–60 μm porosity) contained 4 g of sample achieving a ratio of 16 mg of sample per cm^2^. Two replicates (bags) by incubation period were introduced in a fistulated cow (Holstein breed in a state of maintenance) in reverse order to remove all the bags from the rumen at the same time. Incubation times corresponded to 0, 2, 4, 8, 10, 14, 24, and 48, and 72 h were used. The cow was fed a ration formulated according to its nutritional requirements and managed in the EEAA housing yard. Prior to ruminal incubation, the bags were placed in 20 × 30 cm porous laundry bags and soaked in warm water (30 °C) for 20 min and then introduced into the donor’s cow rumen. Once removed from the rumen, bags were washed with running water until the water was clear and then frozen at −20 °C for 24 h to stop any fermentation activity. Thereafter, samples were thawed and washed in a conventional washing machine for 10 min and dried in a forced-air oven at 60 °C for 48 h. The time 0 h was not incubated in the rumen and was used to determine the soluble fraction of the DM and CP. Residues after ruminal incubation were weighed to determine the amount of feed degraded at a certain incubation time.

Ruminal degradation parameters of OM and CP were determined using the model proposed by Ørskov and McDonald [28], through the nonlinear procedure of the GraphPad Prism v6.0 software, using the exponential model without a lag phase [28]:
PD = A + B (1 − e^−kt^)(1)
where *A* is the soluble fraction (g kg^−1^ fraction of bags washed at time 0 h), *B* is the insoluble but potentially degradable fraction (g kg^−1^), *k* is the degradation rate constant (% h^−1^), and *t* is the incubation time (h).

Effective degradability (ED) was calculated from the aforementioned parameters assuming fractional passage rates (kp) of 5% h^−1^ according to McDonald [29]:
ED (g kg^−1^) = A + B[*k* / (k + *kp*)]e^−kp^(2)
where *A* (g kg^−1^) is the soluble fraction; *B* (g kg^−1^) is the insoluble but potentially degradable fraction; *k* and *kp* (h^−1^) are the ruminal degradation rate constant and passage rate constant (5%), respectively, according to AFRC [30].

A correction for small particle losses was made according to Hvelplund and Weisbjerg [31]. Thus, samples of each food (1.5 g) were placed in containers to which 40 mL of water was added and stored at room temperature (20 °C) for an hour. Then they were filtered and washed eight times with 20 mL of water; the residue was dried in a forced-air oven at 60 °C for 48 h, and then weighed, calculating the water soluble (SOL) fraction. Assuming that, losses of small particles are degraded similarly to the particles left in the bag, corrections can be made for the loss:
*PDcor (ti)* = PD (ti) − P [1 − ((PD (*ti*) − (P + SOL)) / (1 − (P + SOL)))](3)
*EDcor* = SOL + [((1 − SOL) / (1 − (P + SOL))) × (ED − (P + SOL))](4)
acor = A − P(5)
bcor = B + P [B / (1 − (P + SOL))](6)
ccor = c(7)
where: *PDcor (ti)* is the degradability corrected at the time of incubation *ti*, *EDcor* is the corrected effective degradation, *P* are the losses of small particles, and *SOL* is the water solubility.

The values of the in-situ degradation characteristics of OM and CP are shown in Appendix A, which was mainly used for the formulation of treatments and to determine the SI. The adjusted degradation curves of the OM and CP are shown in Appendix A.

### 2.4. In Vitro Fermentation

In vitro gas production was evaluated according to Theodorou et al. [32]. Three replicates (120 mL bottles) were used, including two “control” bottles (a standard concentrate with a known gas production pattern to compare and ensure the proper fermentation process) and two blanks (bottles without substrate) in three incubation runs. One gram of dried lyophilized substrate was used in each bottle. Then, they were filled with 85 mL of Goering–Van Soest medium, gasified with CO_2_, and closed with rubber stoppers, and left at 4 °C overnight. The next day, 4 mL of reducing agent (Distilled water, sodium sulfate, cysteine HCl, and 1 mol/L NaOH) were added; each bottle was gasified again, completely closed with the rubber stopper, and sealed with aluminum. Then they were placed in a water bath at 39 °C. Rumen fluid was obtained directly from heifers in the experiment, randomly selected from the two treatments of each block. The extraction of ruminal liquor was conducted in the morning before heifers were fed, using an oro-ruminal probe (FLORA Ruminator; Profs-Products, Guelph, ON, Canada), stored immediately in thermos flasks and transported to the laboratory. The ruminal liquid was filtered through a cheesecloth, mixed and placed under constant CO_2_ gasification. Later, ruminal liquor from heifers fed their corresponding diet was added (10 mL) to bottles containing the same diet. Once the rumen fluid was inoculated, the initial gas was extracted from the bottles. After inoculation, the bottles were placed in a water bath at 39 °C under continuous horizontal movement at 50 rpm. The gas pressure in the headspace of the bottles, above atmospheric pressure, was measured manually with a pressure transducer (PCE Instruments, Tobarra, Albacete, Spain) at 1, 2, 3, 4, 5, 6, 8, 10, 12, 15, 18, 24, 36, and 48 h, and the volume of gas produced was measured by extraction, using syringes connected through a three-way Luer valve from the bottles until the visual display of the transducer read zero. Once the volume of gas produced was recorded, it was eliminated. Fermentations were stopped after 48 h by placing the bottles on ice. For the in vitro gas production kinetics, after correcting for the white gas production, it was adjusted by the Michaelis–Menten model without lag phase [33] to obtain the corresponding fermentation parameters of the treatments and each of the feeds.
*GP* = *A* × [*T*^n^ / (*T*^n^ + *K*^n^*)]*(8)
where *GP* is the production of gas at time *T*; *A* is the volume of asymptotic gas (mL g^−1^ MS); n is the coefficient that determines the shape of the curve in the function, and *K* is the time in which half of *A* (h^−1^) occurs. The other parameters were calculated according to Groot et al. [34] and France et al. [33]:
*C* = *n* / (2 × *K*)(9)
*MDR* = (*n* − 1) ((*n* − 1) / *n*) / *K*(10)
*ta*, *tb* and *tc* = *K* × (((*X* / (1 − *X*)) (1 / *n*))(11)
where *C* is the degradation rate in the middle of the asymptote; *MDR* is the maximum degradation rate; *ta, tb,* and *tc* (h^−1^) correspond to the time at which 25, 75, and 90% of *A* occurs; and *X* varies between 0.25, 0.75, and 0.90.

Volatile fatty acids (VFA) and ammonium (NH_3_) concentrations were measured only for diets. For this, samples were obtained from the bottles at 4 and 48 h post-incubation, plus the two samples of inoculum to calculate net VFA and NH_3_ production. Then, 5 mL was extracted from the supernatant, and 0.1 mL of concentrated hydrochloric acid (37%) were added. These were stored frozen (−20 °C) and were subsequently analyzed to determine the concentration (mmol L^−1^) and proportion (%) of VFA (acetic acid (C2), propionic acid (C3), butyric acid (C4), isobutyric, valeric and isovaleric [VFAscr]), and the concentration of NH_3_ (mg L^−1^).

### 2.5. The Concentration of N in Feces, Purine and N Derivatives in Urine

A total of 12 heifers (three per pen) were randomly selected to collect fecal samples during the experiment, but one of them was dismissed from the study because of pregnancy. Samples were obtained three times during the experimental period (one day after the 1st, 3rd, and 4th weighing) and always from the same animals. The fecal samples were collected manually from the rectum, then frozen at −20 °C and then lyophilized. The determination of the concentration of N was made with the LECO FP528 nitrogen analyzer described above.

Additionally, urine samples were also obtained from the same 11 heifers by manual stimulation of the vulva. Urine was collected in a 1.0 L container covered with a mesh of lingerie to avoid solid remains coming from around the vulva. A minimum volume of 60 mL of urine was collected for the analysis of the N concentration, plus another 20 mL to determine purine derivatives in the urine. Previously, each bottle was filled with H_2_SO_4_ (10% v v^−1^) of the collected volume. Samples were stored in a cooled container, then frozen at −20 °C. Samples destined to obtain N were lyophilized and subsequently analyzed through the LECO FP528 nitrogen analyzer. Samples for purine derivatives analyses (n = 10) were thawed and analyzed by using the HPLC-UV technique described by Vlassa et al. [35]. Urine volume was estimated according to Al-Khalidi [36], whereas daily total creatinine excretion was calculated according to Chizzotti [37]. Daily excretion of purine derivatives was estimated with the equation of Faichney et al. [38]. The creatinine daily excretion coefficient (mg d^−1^ K) = 113.12 × LW^−0.25^ used was based on Ørskov et al. [39]. Daily purine absorption and the contribution of microbial N were calculated according to Chen and Gomes [40]. Finally, the total microbial protein (TMP) was obtained by the following equation:
TMP = MN × 6.25(12)
where TMP: Total microbial protein (g d^−1^); MN: Microbial Nitrogen (g d^−1^); and 6.25 is the N content in proteins. These results were utilized to estimate the daily synthesis of microbial protein.

### 2.6. Nitrogen Use Efficiency (NUE)

Calculation of NUE was made based on the results of the N balance according to the BCNRM software v1.0.37.14. [24] and the equations proposed by Cole et al. [41] to estimate the retained N based on ADG and LW. The software estimates the amount of N excreted in the urine and feces separately, in addition to the N retained by the animal. Likewise, it estimates the N intake from the diet CP, considering 16% of N. The absorbed N was estimated according to the amount of MP, and the N retained according to the net growth protein. In addition, the software considers both excretions and endogenous N from the animal. Estimations of NUE were made for two periods (Period 1 = first 43 d, and Period 2 = the last 21 d) since consumption was different during these periods. Therefore, the nitrogen consumed was also different at each stage.
NUE = (N retained / N intake) × 100(13)

### 2.7. Carcass Characteristics

Heifers were transported into a slaughterhouse for processing, where the following data were collected: cold carcass weight, carcass pH (measured directly 24 h postmortem), backfat thickness, ribeye area (measured in the cross-section of the *Longissimus dorsi* muscle, between the 9th and 10th ribs), and marbling (on the surface of the 9th rib muscle, determined by using USDA standards.

### 2.8. Data Analysis

The experimental model corresponded to a randomized complete block design (live weight as a block factor) with a univariate treatment structure with two levels (diets). For performance and carcass quality, each animal was considered as an experimental and observational unit. For variables derived from purine in urine, there were 10 observational units (four PCD and six MPD) for the variables of N in urine and feces n = 11 (five CPD and six MPD). The ANOVA model was
Y_ijk_ = μ + β_j_ + α_j_ + ε_ijk_(14)
where μ corresponds to the general mean; βi represents the effect of the i^th^ block; α_j_ represents the effect of the j^th^ diet, and ε_ijk_ the experimental error associated to the k^th^ animal of the j^th^ diet in the i^th^ block. For multiple comparisons, the Tukey test was conducted with a significance = 0.05 when corresponded. In the case of the microbial protein synthesis, N in urine and feces, and results of in vitro fermentation, a repeated measure in the time model was used (Y_ijk_ = μ + α_i_ + S_k(i)_ + β_j_ + [αβ_ij_] + [β × S_jk(i)_] + ε_ijk_, where Y_ijk_ is the observation of the *i*^th^ level diet and the *j*^th^ level of time for subject k; μ is the general mean, α_i_ is the fixed effect level i of diet, S_k(i)_ is the effect of subject nested within a block, β_j_ is the fixed effect of level j of factor time, αβ_ij_ is the interaction effect between diet and time, ε_ijk_ is the random error effect). Since there was no replication for each combination of subject and factor time, the [β × S] interaction effect cannot be separated from the error term and must be assumed to be zero. Animal performance and carcass data were analyzed with the statistical package JMP v14.0 (SAS Institute Inc., Cary, NC, USA). Repeated measures variables were analyzed with the SAS statistical package version 9.4 PROC MIXED (SAS Institute Inc.).

## 3. Results

### 3.1. Animal Performance and Carcass Characteristics

No diet effects for final LW, DMI, ADG, and Feed Conversion (Table 3) were observed. Likewise, no diet × time interaction was observed for ADG (*p =* 0.89), but there was an effect of the time on ADG (*p ≤* 0.001). During the first 21 d, ADG was as expected (near 1.0 kg/d), while in the second period (from 21 to 43 d), there was a slight increase for both diets. Finally, in the last period (43 to 64 d), there was a steep decrease in ADG in both treatments. In addition, no differences (*p* > 0.05) were observed for the weights recorded in the second and third weightings. However, in the last period (43 to 64 d), a decrease in the growth response for both treatments was observed for both the ADG and LW variables.

Heifers fed with the CPD showed greater cold carcass weight (Table 4), approximately 12 kg more than those fed with the MPD diet (*p* = 0.04). However, no differences were observed in the other carcass characteristics (*p* > 0.10), even when the rib eye area was numerically higher in heifers fed with the MPD.

### 3.2. In Vitro Fermentation Products

Parameters of in vitro gas production kinetics for both diets are shown in Table 5. No differences in asymptotic gas and 48 h GP were observed. The MPD exceeded the CPD in the MDR and C, whereas the time to produce 25, 50, 75, and 90% of asymptotic production was greater for CPD than MPD.

The production and proportion of VFAs and NH_3_ concentrations are presented in Table 6. There was a diet by time interaction for the concentration of NH_3_, being similar at 4 h after incubation but at 48 h, the concentrations were 72% higher in the CPD. The total VFA production was greater for CPD when compared to MPD. However, the proportions of propionic acid and VFAscr were higher for MPD, whereas no differences were observed for the proportions of acetic and butyric acids, nor for the acetic: propionic and (acetic + butyric): propionic ratios.

### 3.3. Nitrogen Balance

There was an increase in TMP synthesis over time for both diets (Table 7; *p* < 0.01). In addition, D1, D43, and D64 showed no differences between treatments in the daily amount of TMP produced. Likewise, there was no diet effect on the synthesis of TMP (*p* > 0.05), and the interaction between diet and time was also not significant (*p* > 0.05). There were no differences in time (*p* > 0.05) in the three dates when urinary N was evaluated. The N concentration in the urine was higher in the CPD (*p* < 0.01). Therefore, heifers that received the MPD had a lower concentration of nitrogen in the urine during the fattening period. In addition, there was no interaction between treatments and time (*p* > 0.05).

A significant interaction between diet and time (*p* < 0.05) was observed for the N concentration in the feces, with CPD showing a higher percentage of N over time. Therefore, total N concentrations (urine + feces) of heifers fed a balanced diet in terms of ruminal degradable protein was lower than in those who received an unbalanced ruminal degradable protein diet. Finally, estimation of NUE using the BCNRM software was higher in the MPD (17.44%) when compared to CPD (14.27%) for the whole period. However, when NUE was estimated according to Cole et al. [41], lower values were obtained (8.36% and 9.19%, respectively) decreasing even more in the second period (43 to 64 d), but a without difference between diets (*p* > 0.05).

## 4. Discussion

Nutritional management of ruminants is a relevant issue to a better understanding of the use of nutrients and for optimization purposes, especially in the framework of where environmental, economic, and social matters acquire more relevance [42,43,44,45,46]. The present experiment was conducted to assess the hypothesis that balancing the diet by the MP system would improve N metabolism, rumen fermentation, and microbial protein synthesis in heifers. To test this hypothesis, experimental diets were formulated to have similar SI but differences in RUP and MP balances. To our knowledge, this is the first study that jointly evaluate the SI with the MP system in beef production systems in South America.

In vitro, GP is mainly used to analyze the fermentation kinetics in feed for ruminants [32]. In addition, it has been shown that there is a positive relationship between in vitro GP and digestibility [47]. Although we found no differences in GP, there were differences in VFA production between the two diets (*p* < 0.01). When comparing diets, we observed that CPD produced more total VFA, but that did not reflect a greater production of microbial protein. A possible explanation is that the greater amount of protein was transformed into NH_3_, leaving the carbonated chains that make it up; these would be fermented, producing a greater amount of VFA compared to MPD. However, MPD presented a lower value of the K parameter when compared to CPD, which implies faster fermentation, is in accordance with the MDR being higher for MPD.

On the other hand, increasing CP content in the diet (or N intake) increased the amount of NH_3_ exponentially [3,48]. In addition, the saturation of microbial ammonia uptake has been reported to occur between 10 and 14% CP [49]. Our results agree with these authors, since, an interaction between diet and time was found, with CPD achieving a higher concentration of NH_3_ at 48 h, which would be explained by the higher consumption of N (15.13% vs. 12.73% of CP in CPD and MPD, respectively). Consequently, more N was lost in CPD, which was not used to synthesize microbial protein. In this study, a similar amount of microbial protein was synthesized in both diets, but N in urine (%) was higher in CPD compared to MPD. In this context, Firkins et al. [50] pointed out that there is a higher microbial protein synthesis efficiency when the NH_3_ concentration in the rumen is lower. However, in our study, no differences were observed in the synthesis of microbial protein between diets that showed different concentrations of NH_3_ in the rumen.

Chumpawadee et al. [13] showed that as the SI increases, the greater the synthesis of microbial protein that reaches the small intestine since it seeks to synchronize the source of carbohydrates with N to maximize the efficiency of microbial growth [51]. In our trial, both diets had a similar rate of synchrony, both close to 1.0. This would explain that the synthesis of the microbial protein has not been different between both dietary strategies. In a similar way, the increase in the synthesis of the microbial protein over time could be explained by the same concept. In fact, in the first measurement (day 1), heifers were not yet consuming a synchronized or balanced diet. Thus, as time progressed, the microorganisms adapted to the simultaneous availability of carbohydrates and N sources, which was reflected on measurements done on days 43 and 64. Therefore, microbial growth for both diets was more efficient.

The MP system allows to better adjust the nutrient needs by separating the requirements of the animal from the microbes of the rumen. Consequently, an improvement of some productive parameters and a significant decrease in the excretions of N into the environment is achieved [52]. In this same sense, Henning et al. [9] concluded that the synchrony between energy and the availability of N might be of less importance for microbial growth. However, the energy supply can improve the efficiency of microbial growth. Other authors support this idea [10], since the animal does not have an endogenous system to ensure the supply of energy, but in the case of N, the animal would adapt to ensure the availability of the nutrient through recycling, particularly when it is low in the diet. It has also been reported that at a higher percentage of TDN, which is an indicator of energy in the diet, there is a greater amount of synthesized microbial protein [6]. In this experiment, TDN energy values of both diets were similar (difference of 2.64%), which would confirm the similarity in the synthesis of microbial protein in both dietary strategies.

Seo et al. [53] reported a greater daily microbial growth with higher SI (0.81 and 0.83) when compared with an SI of 0.77. These authors attributed to the fact that their diet with the lower SI had greater amounts of non-structural carbohydrates; therefore, greater production of lactic acid, and consequently, an effect on rumen pH that affects the synthesis of microbial protein [54]. However, in their results, Seo et al. [53] did not report differences in rumen pH. In our study, no differences in pH (data not shown) were observed either, so it follows that it was not a determining factor affecting microbial protein synthesis. Seo et al. [55] suggest that the lower ruminal NH_3_-N concentration correlates with higher utilization of NH_3_-N for microbial protein synthesis. This is in accordance with our study, as MPD had a lower CP concentration and N intake showed a lower concentration of NH_3_ and resulted in similar microbial protein synthesis.

The amount of microbial N produced was within the values described by the literature that used similar breeds [56,57,58], but when compared with breeds destined for milk production, these values were much lower, mainly due to the passage rates compared to those seen in dairy cattle, which were higher and also had higher consumption than beef production cattle [59]. The optimal range of NH_3_ for microbial production is 5 to 8 mg 100 mL^−1^. In our study, NH_3_ was measured in vitro, and for none of the diets, it was limiting microbial growth.

The lowest concentration of N on the three sampling dates was observed in MPD. This would be mainly explained by the balance of the protein through the MP system established by the BCNRM software. The foregoing given that both treatments had a similar SI (0.80 and 0.83), and the difference between them was that it was balanced to adjust the availability of the rumen degradable protein and MP. Meanwhile, CPD was calculated considering the total demand for CP. In our study, a decrease in the total CP concentration of the diet was achieved when it was balanced by the MP system (MPD), reaching 12.73% CP, while the conventional CPD diet was 15.13% CP. Therefore, MPD resulted in a lower excretion of N, as observed in the N concentrations in the urine and feces. Several authors have reported similar results, that is, the higher the content of N consumed, the greater the excretions of N, both for urine and feces [60,61,62]. This is mainly because, by increasing the consumption of CP or rather the RDP, an excess of NH_3_ is produced in the rumen, which is converted to urea in the liver and subsequently eliminated in the urine [63].

On the other hand, Erickson and Klopfenstein [52] found differences in the excretions of N for both one-year-old calves and calves when using a nutritional strategy that incorporates the feeding in phases balanced by the MP system. The authors reported that lower excretions of N were observed in treatments balanced with the MP system. Likewise, as occurred in our study, they pointed out that the contents of CP in the diet balanced by MP were lower. Finally, they reported that adopting this method could reduce excretions in steers in the evaluated fattening period (137 d) by up to 6.1 kg of N. Cole et al. [64] reported similar results regarding the concentration of N in feces and urine, evaluating diets with different CP contents (11.5%, 13.0%, and 14.5%). They observed that the diet with the highest concentration of CP was higher in the N concentration in feces and urine, coinciding with our results.

Carcass characteristics were not affected by the treatments, with the exception of cold carcass weight. This was not expected due to the similarity in the final LW of both dietary treatments. Nevertheless, another study reported that the weight of the hot carcass responded to the increase in the concentration of PC in the diet [65]. On the other hand, Cole et al. [64] and Gleghorn et al. [66] reported a lack of differences in the degree of marbling and fat coverage regarding the change of protein offered in the diet. In contrast, there was a trend for a smaller ribeye area in diets with higher CP, that is, 14% vs. 12% (Cole et al. 2003). Nevertheless, our results must be analyzed with caution because of the low number of replicates in the study.

The values of the NUE herein reported are within the ranges reported in the literature [21,41,45]. The N consumed in MPD was lower compared to CPD, because CP content was lower, and the RUP balance was better. The latter resulted in a lower concentration of NH_3_ in MPD, and when there is less ruminal NH_3_, NUE use to be higher [54]. In addition, the concentration of N in the feces and urine was lower in heifers fed with MPD. Therefore, they consumed less N and utilized it more efficiently than heifers fed the CPD. Although both, MP system and the SI, seek to improve NUE [12,44], using them together can produce better results than using each one separately. Galyean [67] promotes the use of the MP system since the protein intake is calculated according to the requirements of the microorganisms, thus avoiding excesses of N in the diets and fewer excretions of N into the environment. Therefore, using the MP system can reduce N excretions and therefore improve NUE [44,45].

By using the equations of Cole et al. [41], a numerically lower NUE was obtained in both periods but showing the same behavior as the values obtained from the BCNRM software. The lower NUE observed in the second period was due to animals decreasing their ADG caused by the decrease in DMI. Therefore, they were less efficient in retaining the ingested N. This scenario was not reflected with the values obtained using the BCNRM software.

Although there is no official data in Chile, some reports [68], as well as the empirical experience, indicate values of CP ranging from 10.5 to 20.0%, whereas energy metabolizable energy (ME) range between 2.60 to 2.90 Mcal/kg DM. This variation is explained in part because many production systems, particularly in Southern Chile, are grass-fed and use strategic supplementation with concentrates (usually small cereal grains and byproducts). Meanwhile, feed yards are closer to 13.5% CP and 2.75 Mcal ME/kg DM, but still showing great variability among them. Ruminal protein balance (RPB) has been proposed as a new trait to be used in diet formulation by Sauvant and Nozière [69] in the INRA (Institute National de Recherche Agronomique) 2018 system because it induces several interesting responses of protein and energy efficiency. These authors also indicated a negative relation of RPB with microbial growth efficiency and highlighted the relevance of degradable CP and energy in the rumen. Also, it has been demonstrated a decrease of OM digestibility when RPB decrease under zero [69,70].

Most of the actuals systems (NASEM, INRA 2018, DVE/OEB, NorFor, AFRC, CSIRO) have a mechanistic approach that looks at improving animal performance, product quality, the efficiency of feed use, and at the same time considers the environmental impacts by reducing N excretion and CH_4_ emissions. The last release of INRA system (2018), has considered as a relevant criterion not only to evaluate the balance between degradable N and the energy available in the rumen but also to integrate the quantitative effects of the energy × nitrogen interactions in digestive processes, as previously described by Sauvant and Nozière [69]. A major focus in the new model was fractional passage rates of liquids, forages, and concentrates (proportion of concentrate in the diet), as well as the level of feeding. In our case, we use a fractional passage rate constant of 0.05%/h, but based on equations of the INRA 2018, it should be 0.04%/h. This, as mentioned before, affected the digestibility of the OM and the results associated with the NUE.

## 5. Conclusions

A dietary strategy that combines a high synchrony index with the MP system increases the NUE without negatively affecting animal performance or carcass characteristics compared to heifers fed a diet formulated based on their CP requirements. The lower N excretions show that the MPD is more environmentally friendly. In addition, the lower CP content in the MP diet may increase the beef production system’s profitability because protein is one of the highest costs in the ration. Finally, we accepted the hypothesis, since the combination of synchrony and the MP system achieve greater efficiency in the use of nitrogen, without negatively affecting animals’ performance neither the quality of the carcass.

## Figures and Tables

**Table 1 animals-10-00852-t001:** Nutritional characteristics of the ingredients utilized in the study.

Item	Pasture Haylage *	Canola Meal	Sugar Beet Pulp	Triticale Grain
Dry Matter, %	49.67	91.70	88.90	90.35
Crude Protein, %	14.31	33.48	8.69	8.13
Soluble Protein, %	8.10	8.56	0.17	1.79
Neutral Detergent Fiber, %	53.68	32.15	50.22	20.32
Acid Detergent Fiber, %	30.58	21.51	24.65	3.79
DOMD, %	76.20	74.40	90.12	92.52
TDN, %	76.50	74.83	88.97	91.17
Organic matter, %	92.98	94.19	94.75	98.40
NFC, %	21.34	16.97	35.11	68.43
Ether Extract, %	3.65	11.59	0.73	1.52
Ash, %	7.02	5.81	5.25	1.60
ME, Mcal kg^−1^ DM	2.76	2.70	3.21	3.29
NEm, Mcal kg^−1^ DM	1.83	1.78	2.20	2.27
NEg, Mcal kg^−1^ DM	1.20	1.16	1.52	1.57

* Purchased to local farmers, made mainly of *Lolium spp*. but also included a lower proportion of *Dactylis glomerata*, *Bromus spp*., and *Trifolium spp*. DOMD = digestible organic matter on a dry matter basis; TDN = Total Digestible Nutrients estimated as ((ME/0.82)/4.4) × 100; NFC = Non-fiber carbohydrates estimated as 100—(%NDF + %CP + %Ash + %EE); ME = Metabolizable energy; NEm = Net energy of maintenance, and NEg = Net energy of gain.

**Table 2 animals-10-00852-t002:** Nutritional characteristics of the diets, ingredient inclusion, and synchronize index per treatment.

Item	Treatment	
	CPC	MPD
Synchrony index (SI) ^†^	0.80	0.83
Ingredient inclusion in the diet		
Pasture haylage, %	50.0	58. 0
Canola meal, %	15.0	4.0
Sugar beet pulp, %	20.0	0.0
Triticale grain, %	15.0	38.0
Dry Matter, %	64.1	61.3
Crude Protein, %	15.1	12.7
Soluble Protein, %	5.6	5.7
Neutral Detergent Fiber, %	44.8	40.1
Acid Detergent Fiber, %	24.0	21.5
Digestibility value, %	81.2	74.4
TDN, %	66.2	68.8
Organic Matter, %	94.3	95.1
NFC, %	30.5	39.1
Ether Extract, %	3.9	3.2
ME, Mcal kg^−1^ DM	2.9	3.0
NEm, Mcal kg^−1^ DM	2.0	2.0
NEg, Mcal kg^−1^ DM	1.3	1.4
RDP, (% CP) ^‡^	86.0	83.3
RUP, (% CP) ^‡^	14.0	16.8
MPB, (g/d) ^‡^	5.8	0.1
RDPB, (g/d) ^‡^	240.0	−10.0

CPD: Crude protein diet; MPD: metabolizable protein diet; TDN = Total Digestible Nutrients estimated as ((ME/0.82)/4.4) × 100; NFC = Non-fiber carbohydrates estimated as 100—(% NDF + % CP + % Ash + % EE); ME = Metabolizable energy; NEm = Net energy of maintenance, and NEg = Net energy of gain; RDP = Ruminal degradable protein; RUP = Ruminal undegradable protein; MPB: Metabolizable protein balance; RDPB = Ruminal degradable protein balance. ^†^ Calculated from in situ degradation parameters and Verbič et al. [23] equations. ^‡^ Data obtained from the BCNRM software v1.0.37.14. [24]. Both diets were supplemented with 80 g d^−1^ per heifer of mineral salts.

**Table 3 animals-10-00852-t003:** Least square means for live weight, dry matter consumption, feed conversion, and daily weight gain per diet.

Variable	CPD ± SEM	MPD ± SEM	*p*-Value
Initial LW, kg	390.7 ± 6.09	386.6 ± 5.51	0.302
Final LW, kg	445.9 ± 4.72	437.5 ± 4.33	0.207
ADG, kg d^−1^			
d 0–21	1.01 ± 0.15	0.90 ± 0.14	0.604
d 21–43	1.24 ± 0.15	1.02 ± 0.14	0.308
d 43–64	0.36 ± 0.15	0.30 ± 0.14	0.774
d 0–43	1.11 ± 0.10	0.95 ± 0.09	0.234
d 0–64	0.87 ± 0.07	0.74 ± 0.06	0.200
DMI, kg d^−1^	8.56 ± 0.04	8.49 ± 0.03	0.192
Feed conversion rate, kg DM kg LW gain ^−1^	10.7 ± 1.42	12.3 ± 1.59	0.381

CPD: Crude Protein Diet; MPD: Metabolizable Protein Diet; ADG = Average daily gain; DMI = Dry matter intake; LW = Live body weight.

**Table 4 animals-10-00852-t004:** Least square means for carcass characteristics variables measured in heifers fed with two different diets.

Variable	CPD	MPD	*p*-Value
Cold carcass weight, kg ^†^	240.2 ± 4.12 ^a^	228.3 ± 3.55 ^b^	0.041
Ribeye area, cm^2^	88.1 ± 3.10	92.6 ± 2.68	0.281
Back fat thickness, mm	7.0 ± 1.10	6.5 ± 0.91	0.704
Marbling ^‡^	1.9 ± 0.20	1.8 ± 0.17	0.728
Muscle pH	5.0 ± 0.10	5.1 ± 0.05	0.135
Dressing, %	54.0	53.0	0.150

CPD: Crude Protein Diet; MPD: Metabolizable Protein Diet; ^†^ Different letters between columns indicate statistical difference, Tukey test; ^‡^ Marbling values scale from 1 to 3 (1 = Choice +; 2 = Choice; and 3 = Choice −).

**Table 5 animals-10-00852-t005:** Least square means of the parameters of the in vitro gas production kinetics of diets.

Variable	Diets		SEM	*p*-Value
	CPD	MPD		
GP48	248.6	257.4	9.33	0.125
A	266.7	269.1	10.84	0.715
n	1.8	1.8	0.06	0.128
K	12.2	11.1	0.42	0.008
C	0.1	0.1	0	0.003
MDR	0.1	0.1	0	0.004
ta	6.6	6.1	0.36	0.039
tb	22.6	20.2	0.44	0.004
tc	42.1	36.6	1.22	0.004

CPD = Crude Protein Diet; MPD = Metabolizable Protein Diet; GP48: Gas volume produced at 48 h of incubation (mL g^−1^ DM); A = Asymptote gas volume (mL g^−1^ DM); n = coefficient that determines the curve in the function; K = time to produce 50% of A (h), C: fractional rate of gas production in the middle of the asymptote (h^−1^), MDR = maximum degradation rate (h^−1^); ta, tb y tc = time to produce 0.25, 0.75 y 0.90 of A (h^−1^).

**Table 6 animals-10-00852-t006:** Least square means for the production and concentration of volatile fatty acids (VFAs) and the concentration of ammonium (NH_3_) for both diets.

Variable	Diets			Time			*p*-Values		
	CPD	MPD	SEM	4 h	48 h	SEM	Diet	Time	D × T
NH_3,_ mg L^−1^	89.9	66.5	4.6	67.8	88.6	4.48	<0.05	<0.01	<0.01
Total VFA, mmol L^−1^	21.0	15.8	0.9	4.1	32.7	1.28	<0.01	<0.01	NS
Acetate, mol 100 mol^−1^	58.0	52.4	2.5	63.6	46.8	2.83	NS	<0.05	NS
Propionate, mol 100 mol^−1^	29.8	32.9	1.9	27.0	35.7	2.27	<0.05	<0.05	NS
Butyrate, mol 100 mol^−1^	9.9	11.3	2.1	8.9	12.3	1.84	NS	NS	NS
VFAscr, mol 100 mol^−1^	2.3	3.5	0.1	0.6	5.2	0.25	<0.01	<0.01	NS
C2:C3	2.1	1.7	0.2	2.5	1.3	0.29	NS	<0.05	NS
(C2 + C4):C3	2.4	2.1	0.3	2.8	1.7	0.33	NS	<0.05	NS

CPD = Crude Protein Diet; MPD = Metabolizable Protein Diet; SEM = Standard error of the mean; NH_3_ = ammonia concentration; VFAscr = Branched chain volatile fatty acids production (isobutiric + valeric + isovaleric); C2:C3 = ratio acetic: propionic acid; (C2 + C4)/C3 = ratio ketogenic acids: glucogenic acid.

**Table 7 animals-10-00852-t007:** Least square means for microbial protein, purine derivatives, creatinine, urine, and feces nitrogen concentration for both diets.

Variable	Diets				Time				*p*-Values		
	CPD	SEM	MPD	SEM	Day 1	Day 43	Day 64	SEM	Diet	Time	D × T
TMP, g d^−1^	581.2	47.8	643.4	52.3	420.4	670.3	746.2	44.6	NS	<0.01	NS
N_Mic_, g d^−1^	93	7.6	102.9	8.4	67.3	197.2	119.4	7.1	NS	<0.01	NS
A, mmol L^−1^	11.4	2.2	13.9	2.4	14.2	11.2	12.6	2.2	NS	NS	NS
UA, mmol L^−1^	1.7	0.3	2.0	0.3	2.6	2.3	0.5	0.2	NS	<0.01	NS
C, mmol L^−1^	11.5	2.1	14.7	2.3	20.1	9.9	9.2	2.1	NS	<0.01	NS
TPD, mmol L^−1^	13.0	2.4	15.9	2.6	16.8	13.5	13.1	2.3	NS	NS	NS
N in urine, %	7.4	0.4	4.9	0.3	6.1	6.5	6.3	0.5	<0.01	NS	NS
N in feces, %	3.3	0.1	2.7	0.1	2.9	2.8	3.0	0.1	<0.01	NS	<0.05

CPD = Crude Protein Diet; MPD = Metabolizable Protein Diet; SEM = Standard error of the mean; TMP = Total microbial protein; N_Mic_ = Microbial nitrogen; A = Allantoin; UA = Uric acid; C = Creatinine; TDP = Total purine derivatives (allantoin + uric acid); N urine = Concentration of nitrogen in urine; N feces = Nitrogen concentration in feces; NS = Not significant.

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
