# Peer review of "Effect of Two Nutritional Strategies to Balance Energy and Protein Supply in Fattening Heifers on Performance, Ruminal Metabolism, and Carcass Characteristics"

_animals, 2020, doi:10.3390/ani10050852_

Round 1

Reviewer 1 Report

I had reviewed the manuscript. I had attached my comments in the file, please find it. 

Author Response

Please, see the attached file

Reviewer 2 Report

This study evaluated if a diet formulated with the MP system and synchronized (energy: protein) would decrease the environmental impact by reducing nitrogen losses to the environment, without negatively affecting in vitro rumen fermentation, the productive performance and the carcass characteristics of the fattening heifers. As you mention in Table 2, the protein parameters of MP system (RDP, RUP, MPB and RDPB) are predicted but not analysed. In fact, the only measured parameter was CP, and the reader could infer that this was the key difference between diets. The reader cannot see the differences between diets in OM and CP degradation curves (not shown in suppl. Files). In addition, I wonder why the digestibility value was lower if the fibre fractions were also lower in the MPD compared to CPC. The N outcomes in urine and feces support that CP (and energy:CP ratio) was the main difference between the diets (and of course accordingly MP), but then there would be no need to dizzy the data with many estimated calculations for MP parameters (the reader could estimate them on their own, if necessary, based on the analysed parameters that you present). Please summarize the main parameters that are necessary to support your conclusions but avoid all the calculations regarding MP system. L112 humane animal care? L121-122 if age of heifers cannot be assured, provide an approximate percentage of their attained mature weight at the start of the experiment L126 space allowance per animal rather than total space available. L168 which pasture haylage species? L207-227 You did not show in supplementary material the OM and CP degradation curves but the ingredient degradation curves. How can you affirm that both diets (after mixing the ingredients) differed in these parameters? L271 You said first in L125 that 24 heifers were located in two pens. Now you said that you selected 12 heifers (three per pen?). Clarify. L348-349 Why did not using jmp for repeated measurements mixed model? (instead of sas) Table 3 and 4, a single decimal would be enough…as the weighing system cannot provide further precision than 100 g, and similar for mm… Table 6, why do you express VFA both in mmol/L and %. Why don’t you use only the most extended form? Are both of them necessary to support your results? L428-430 how do you explain that no differences in nitrogen use efficiency between diets were observed? L534 some discussion regarding the lack of differences in carcass composition would be required. A discussion with the recommended dietary CP (based on different nutrient requirement models as NRC or INRA) for the Hereford x Angus heifer crossbred would be required. Which is the dietary CP and energy that is formulated by the farming industry to feed this animal genotype in Chile or surrounding countries?

Author Response

Please, see the attached file

Round 2

Reviewer 2 Report

My suggestions were mostly addressed. I would prefer not using two decimals in some units because weight precision cannot be assured up to this point (for example body-weight, or some rumen fermentation parameters). For example, did you use a cattle weighing machine with more than 100 g precision? This is independent to using least square means or not (you can use a least square mean with only a single decimal).

I wonder if autor contribution style fits Journal standards...

Author Response

We have followed the indications of the reviewer regarding the number of decimals. Changes have been made on Table 3 (initial and final body weight and feed conversion rate), as well as in Tables 2, 4, 5, and 6 (in those cases all values have been left with only one decimal).

The reviewer can find these changes with the tracking systems of Word.
